# Learning Clinically Acceptable Segmentation of Organs at Risk in Cervical Cancer Radiation Treatment from Clinically Available Annotations

**Monika Grewal**[1]                                                    MONIKA.GREWAL@CWI.NL

[1] *Centrum Wiskunde & Informatica, Amsterdam, The Netherlands*

**Dustin van Weersel**[2]                                              DUSTIN@XOMNIA.COM

[2] *Xomnia B.V., Amsterdam, The Netherlands*

**Henrike Westerveld**[3]                                             G.WESTERVELD@ERASMUSMC.NL

[3] *Erasmus University Medical Center, Rotterdam, The Netherlands*

**Peter A. N. Bosman**[1,4]                                            PETER.BOSMAN@CWI.NL

[1] *Centrum Wiskunde & Informatica, Amsterdam, The Netherlands*

[4] *Delft University of Technology, Delft, The Netherlands*

**Tanja Alderliesten**[5]                                             T.ALDERLIESTEN@LUMC.NL

[5] *Leiden University Medical Center, Leiden, The Netherlands*

**Editors:** Accepted for publication at MIDL 2023

## Abstract

Deep learning models benefit from training with a large dataset (labeled or unlabeled). Following this motivation, we present an approach to learn a deep learning model for the automatic segmentation of Organs at Risk (OARs) in cervical cancer radiation treatment from a large clinically available dataset of Computed Tomography (CT) scans containing data inhomogeneity, label noise, and missing annotations. We employ simple heuristics for automatic data cleaning to minimize data inhomogeneity and label noise. Further, we develop a semi-supervised learning approach utilizing a teacher-student setup, annotation imputation, and uncertainty-guided training to learn in presence of missing annotations. Our experimental results show that learning from a large dataset with our approach yields a significant improvement in the test performance despite missing annotations in the data. Further, the contours generated from the segmentation masks predicted by our model are found to be equally clinically acceptable as manually generated contours.

**Keywords:** organs at risk, segmentation, deep learning, missing annotations

## 1. Introduction

The planning for cervical cancer radiation treatment[1] requires manual contouring of the Organs at Risk (OARs) where the adverse effects of radiation must be minimized. Automatic segmentation of these OARs can save hours of manual work. In this paper, we focus on the automatic segmentation of four OARs in cervical cancer radiation treatment: bowel bag, bladder, hips, and rectum. A few studies have focused on developing deep learning based automatic OARs segmentation methods for cervical cancer radiation treatment (Liu et al., 2020b,a; Wang et al., 2020; Mohammadi et al., 2021; Rigaud et al., 2021). All of these studies use a traditional setup for developing a deep learning model, which involves:

---

1. Radiation treatment for cancer involves giving high doses of radiation to the tumor to kill cancer cells.

(a) obtaining a fully annotated clinically available dataset, (b) splitting the data into training, validation, and testing, and (c) training a model and evaluating it on the test dataset. A major drawback in this setup is the limited size of the datasets used for training and testing. A small training dataset limits the possibility of a deep learning model capturing large variance in real-world data. Further, evaluation results from a small test dataset do not inform sufficiently in regard to the true test performance of a deep learning model. Although in the medical imaging domain, such a setup is understandable because of the underlying requirement of clinical expertise for annotating the data, it would be of interest to investigate if clinically available data can be leveraged to increase the size of the training and testing datasets.

The size of the training dataset for automatic OARs segmentation for cervical cancer radiation treatment can be increased if the abdominal scans acquired for tumors other than cervical cancer are also included. However, all the OARs in cervical cancer radiation treatment may not be annotated in those scans. Furthermore, since the clinically available abdominal scans and annotations are retrospectively included, the acquisition protocols, contouring guidelines, and observers may be different giving rise to data inhomogeneity and label noise. In this paper, we follow the motivation of harnessing the benefits of training on a large dataset. Therefore, we use the Computed Tomography (CT) scans and OARs contours delineated for clinical use during radiation treatment for tumors in the abdominal region to develop a deep learning model for segmentation of OARs in cervical cancer radiation treatment. We develop a semi-supervised learning approach to tackle the issue of missing annotations in data. Briefly, the key contributions of our work are the following:

1. We propose a teacher-student setup, wherein, the predictions from a teacher model are used to impute the missing annotations, and a student model is trained using the dataset containing imputed annotations. Additionally, we train the student with an uncertainty-guided loss to avoid the adverse effect of imperfect predictions from the teacher, and with additional augmentations to increase performance.
2. We perform an ablation study to investigate the effect of different components of the proposed approach. Furthermore, we perform a clinical validation study to assess the clinical acceptability of contours generated from automatic segmentation masks predicted by our deep learning model.

### 1.1. Related Work

Our approach is closely related to previous works in the direction of semi-supervised learning by generation of pseudo-labels and self-training for medical image segmentation tasks (Bai et al., 2017; Li et al., 2019, 2020; Zheng et al., 2020). Different from these works, we use self-training with pseudo-labels in a teacher-student setup similar to (Sedai et al., 2019; Yu et al., 2019). Further, we utilize uncertainty maps to reduce the adverse effect of imperfect pseudo-labels, which have been previously used in (Sedai et al., 2019; Yu et al., 2019; Zheng et al., 2020). In contrast to (Sedai et al., 2019; Yu et al., 2019), we train a noisy student with the use of additional augmentations in the data because it has been shown to provide performance gain (Xie et al., 2020). In the domain of learning an OARs segmentation model for cervical cancer radiation therapy by utilizing a large dataset, our work is similar to (Rhee et al., 2020). However, instead of learning a separate model for each OAR as

in (Rhee et al., 2020), we learn a single model for the segmentation of all OARs, which increases the potential for real-world deployment of our model.

## 2. Data

We retrospectively selected the CT scans of female patients who were treated in an academic hospital for a tumor in the abdominal region from 2009 to 2019. A total of 1170 CT scans with associated clinically available contours from 1108 patients were received in anonymized form through a data transfer agreement. These scans were used for training and validation. For testing, we used 105 CT scans with associated clinically available contours from 95 cervical cancer patients who received radiation treatment in the same hospital.

### 2.1. Preprocessing

In all the CT scans (1170 from the training and validation dataset, and 105 from the test dataset), the clinically available annotations of four OARs in cervical cancer radiation treatment (bowel bag, bladder, hips, and rectum) were extracted by using the following steps: (1) standardize different variations of organ labels (e.g., bowel, bowel bag, Bowel bag, bowel_bag, Bowel_bag were all considered bowel bag), (2) combine left and right hip annotations as a single organ, (3) remove voxels annotated as bladder or rectum from the bowel bag annotation to avoid ambiguous labeling in those voxels. Next, the scans were resampled to 2.5mm×2.5mm×2.5mm voxel spacing. The Hounsfield units were converted to intensity values between 0 and 1 by windowing (window level=40, window width=400). In the training and validation dataset, the preprocessing resulted in a total of 186 scans that contained annotations for all the four OARs considered in this work (referred to as the fully annotated dataset, $\mathcal{D}_f$). The remaining scans had missing annotations for at least one of the OARs (referred to as the partially annotated dataset, $\mathcal{D}_p$). In total 383, 1103, 504, and 865 scans had annotations for bowel bag, bladder, hips, and rectum, respectively.

### 2.2. Automatic Data Cleaning

Since the data was accumulated over 10 years and the scans belonging to patients who were treated for a tumor anywhere in the abdominal region were included, the data exhibited inhomogeneity in the cranial extent of the scan (causing an increase in the number of background voxels and potentially less efficient training), and the cranial border of the bowel bag annotations (attributing to label noise).

To make the data more homogeneous so that the adverse effects of inefficient training and label noise could be reduced, we analyzed the histograms of $\mathcal{D}_f$ and decided on thresholds such that the histograms represented a unimodal distribution corresponding to the most frequently used scanning protocol and annotation style (details are provided in appendix A). Based on these thresholds, the scans were cropped in the cranial direction to remove the chest region. The bowel bag annotations in the abdominal region roughly above the level of the lumbar (L4) spinal segment were deleted. The scans that did not contain bowel bag annotations in the entire pelvic region were discarded. These steps resulted in a decrease in the size of $\mathcal{D}_f$ from 186 to 134. The resulting dataset of 134 scans is referred to as $\mathcal{D}_f^{clean}$ in the rest of the paper.

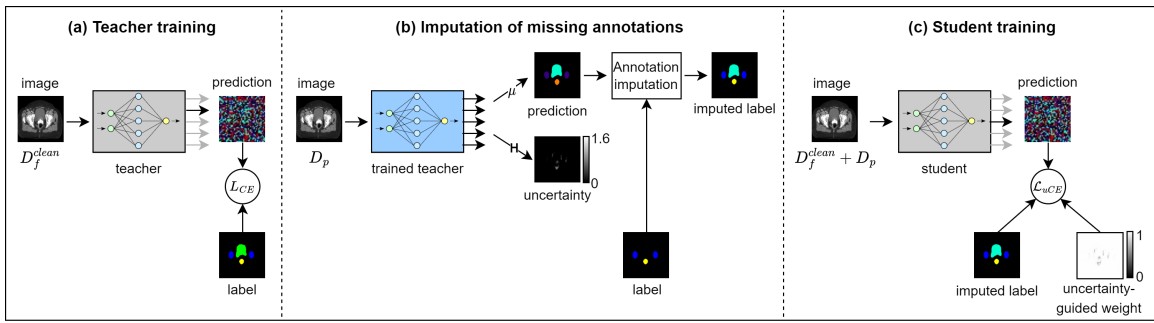

Figure 1: Schematic of the proposed approach. (a) A K-head (depicted by output arrows) teacher model is trained by randomly selecting a single head (highlighted in black) for backpropagation. (b) The clinically available 'label' contains annotation for hips (blue) and rectum (yellow) only. The annotation for bladder is missing. The mean prediction (of K-heads) from the trained teacher is used to impute the bladder annotation. (c) A K-head student model is trained with imputed label and uncertainty-guided loss. $\mu$: mean, H: entropy, $L_{CE}$: cross-entropy loss, $\mathcal{L}_{uCE}$: uncertainty-guided loss.

## 3. Approach

We developed a semi-supervised learning approach utilizing a teacher-student setup (Figure 1). We train a teacher model using the small, fully annotated dataset ($\mathcal{D}_f^{clean}$). The predictions from the trained teacher model are used to impute the remaining large dataset with missing annotations ($\mathcal{D}_p$). Then, a student is trained with the entire dataset ($\mathcal{D}_f^{clean} + \mathcal{D}_p$) containing the clinically available and imputed annotations.

### 3.1. Uncertainty-Guided Training

Epistemic uncertainty refers to the lack of knowledge in a model about the underlying data. Estimating epistemic uncertainty enables the estimation of the reliability of a model's prediction for a specific sample. We train the teacher model to also estimate the epistemic uncertainty maps for each sample. For this purpose, we use a K-head neural network, similar to (Zheng et al., 2020). At each iteration of training, a single head is selected randomly for backpropagation. During inference, we use the mean prediction from K-heads as confidence and the entropy of the mean prediction as an estimate of epistemic uncertainty. We selected the K-head approach because it allows independence between predictions from different heads with faster inference times as compared to the Monte-Carlo (MC) dropout approach (Gal and Ghahramani, 2016). Moreover, the memory overhead is not much compared to fully independent deep ensembles (Lakshminarayanan et al., 2017).

We train the student model with an uncertainty-guided cross-entropy loss $\mathcal{L}_{uCE} = e^{-u}y \cdot log(\hat{y})$, where $u$ is uncertainty in the teacher's predictions at each voxel, $e^{-u}$ is the uncertainty-guided weight, $y$ is the reference label, and $\hat{y}$ is the predicted probability. The weight $e^{-u}$ ensures a large weight on voxels where the uncertainty in the teacher's predictions is small and vice-versa. We set $u = 0$ at the voxels where annotations are clinically available. In this way, the student model can benefit from training with a large dataset while avoiding deterioration in performance due to uncertain label predictions from the teacher model.

### 3.2. Implementation Details

As a baseline, we used the original U-Net architecture (Ronneberger et al., 2015) after replacing the 2D convolutional layers with 3D convolutional layers and adding a batch normalization layer after each convolutional layer. The training was done using randomly cropped 3D patches (of depth 32 along the transverse direction) with a batchsize of 1 because of the GPU memory constraints. The implementation[2] was done in Python by using the PyTorch library (Paszke et al., 2017) and the training was done on NVIDIA RTX2080 GPUs. Other hyperparameters were: optimizer=Adam (Kingma and Ba, 2015); network initialization=Kaiming He (He et al., 2015); learning rate (LR)=$1e^{-3}$; weight decay=$1e^{-4}$; the number of training epochs=500 for teacher models, 250 for student models; learning schedule=step LR with step size=$\frac{1}{3}\times$total training steps; data augmentations=global brightness and contrast variations ($\pm20\%$), random rotations (-10° to 10° along all axes); the number of heads (K) in teacher and student=5.

| Method | Dice (%) | Surface Dice (%) | HD |
|---|---|---|---|
| 3D U-Net + $\mathcal{D}_f$ | 83.47 (6.16) | 80.23 (6.82) | 16.06 (9.07) |
| 3D U-Net + $\mathcal{D}_f^{clean}$ | 85.02 (5.92)* | 82.00 (6.55)* | 12.44 (10.58)* |
| *basic teacher* | 85.36 (5.54)* | 82.33 (6.18)* | 11.61 (7.94)* |
| *basic student* | 87.01 (4.62)*† | 84.64 (5.18)*† | 10.64 (8.00)*† |
| *robust teacher* | 85.31 (5.25)* | 82.30 (5.72)* | 11.57 (7.73)* |
| *basic teacher + robust student* | 87.11 (4.28)*† | 84.76 (4.85)*† | 10.39 (6.68)*† |
| *robust teacher + robust student* | 87.16 (4.19)*† | 84.82 (4.68)*† | 9.92 (4.72)*† |
| *robust teacher + robust student* - iter. 2 | 87.40 (4.13)*† | 85.30 (4.60)*† | 9.85 (4.86)*† |
| *robust teacher + robust student* - iter. 3 | 87.35 (4.10)*† | 85.24 (4.63)*† | 9.96 (4.84)*† |

Table 1: Mean (standard deviation) of mean test performance per scan of the best models obtained from 5-fold cross-validation. Aug.: additional augmentations, HD: Hausdorff distance in mm at 95 percentile. Surface Dice were computed at a tolerance of 2.5mm (voxel spacing). *significant differences compared to 3D U-Net + $\mathcal{D}_f$, †significant differences compared to 3D U-Net + $\mathcal{D}_f^{clean}$.

### 4. Ablation Experiment

We conducted an ablation experiment to look into the individual effect of the components of our approach. As a baseline, we used two models: 3D U-Net trained with $\mathcal{D}_f$, and 3D U-Net trained with $\mathcal{D}_f^{clean}$. Note that the 3D U-Net trained with $\mathcal{D}_f^{clean}$ is similar to the traditional setup of deep learning model development. In the first stage of ablation, we trained a K-head 3D U-Net teacher model with $\mathcal{D}_f^{clean}$ (referred to as '*basic teacher*') followed by K-head 3D U-Net student model with the large dataset ($\mathcal{D}_f^{clean} + \mathcal{D}_p$) and uncertainty-guided loss (referred to as '*basic student*'). In the next stage, we employed the following

---

2. The source code is available at https://github.com/monikagrewal/OrganSegmentation.

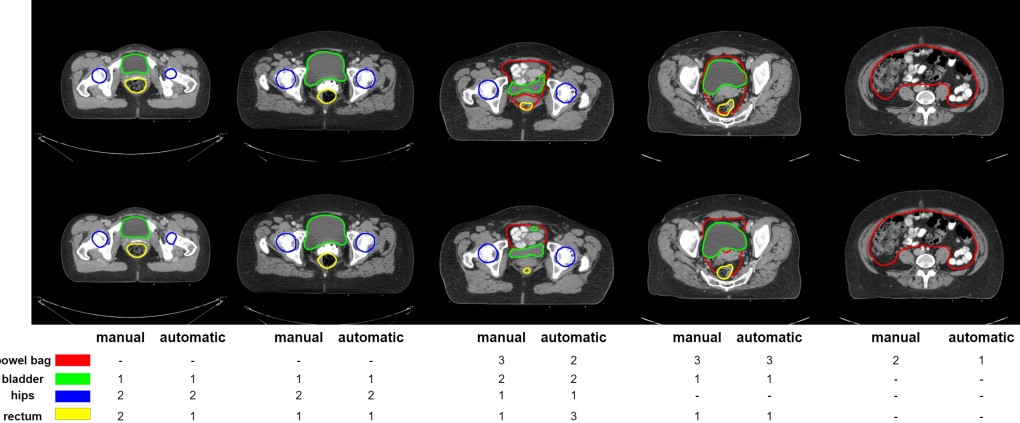

Figure 2: Representative examples of OARs contours. *Top row*: clinically available contours (manual), *Bottom row*: contours generated from OARs segmentation masks predicted by our approach (automatic). Further, the clinical acceptability grades (smaller value indicates better quality) are reported for each OAR.

additional data augmentations to introduce noise in the data: left-right flipping, masking an organ with a random intensity to simulate contrast, global elastic deformations, and elastic deformations centered in either bowel bag or bladder as additional augmentations. We compared the performance of three models: a teacher model trained with $\mathcal{D}_f^{clean}$ and additional augmentations (referred to as '*robust teacher*'), a student model trained with $\mathcal{D}_f^{clean} + \mathcal{D}_p$ and additional augmentation, and using the imputed annotations from *basic teacher* (referred to as '*basic teacher + robust student*'), and a student model trained with $\mathcal{D}_f^{clean} + \mathcal{D}_p$ and additional augmentation, and using the imputed annotations from *robust teacher* (referred to as '*robust teacher + robust student*'). Further, we performed 3 iterations of teacher-student training for *robust teacher + robust student*, wherein in each subsequent iteration, the student model became the teacher and a new student model was trained.

The mean and standard deviations of the performance metrics on test data from the best models obtained after 5-fold cross-validation are reported in Table 1. The distributions of performance metrics for each method (N = 105 test scans × 5 models) were tested for normality using the Kolmogorov-Smirnov test. This was followed by a Friedman test for the main effect and Wilcoxon signed-rank test for post-hoc comparisons. A p-value less than 0.05 with adjustment for multiple comparisons was considered significant.

The automatic data cleaning had a significant impact on the test performance ($p = 5.96e^{-}18$, $p = 6.76e^{-}17$, $p = 2.18e^{-}29$ for Dice, Surface Dice (SD), and Hausdorff distance (HD), respectively), which was mainly due to better bowel bag segmentation. The automatic data cleaning increased the mean Dice coefficient of the bowel bag from 0.7947 to 0.8477 (performance metrics for all the OARs separately are provided in Appendix B). Furthermore, learning from a large dataset with the proposed teacher-student setup, annotation imputation, and uncertainty-guided training (*basic student*) provided a significant gain of 2.34% in mean Dice coefficient ($p = 4.51e^{-}38$), 3.22% in mean SD ($p = 1.21e^{-}35$), and 14.47% in mean HD ($p = 1.51e^{-}15$) as compared to learning from a small, fully annotated dataset (U-Net + $\mathcal{D}_f^{clean}$). Adding noise to the data through additional augmentations provided only a marginal gain in the mean performance of the student model, but a consid-

|                       | A    | B     | C    | D     | E1   | E2   | Ours |
|-----------------------|------|-------|------|-------|------|------|------|
| Bowel bag             | -    | -     | 0.85 | -     | 0.78 | 0.78 | 0.86 |
| Bladder               | 0.91 | 0.92  | 0.91 | 0.89  | 0.90 | 0.91 | 0.92 |
| Hips                  | 0.88 | 0.905 | 0.90 | 0.935 | 0.89 | 0.92 | 0.93 |
| Rectum                | 0.81 | 0.79  | 0.82 | 0.81  | 0.77 | 0.77 | 0.78 |
| Number of test samples | 25   | 14    | 27   | 140   | 30   | 30   | 105  |

Table 2: Mean Dice coefficients reported in A:(Wang et al., 2020), B:(Liu et al., 2020b), C:(Liu et al., 2020a), D:(Rhee et al., 2020), E1:(Rigaud et al., 2021) model 1, E2:(Rigaud et al., 2021) model 2, and Ours: *robust teacher + robust student*.

erable decrease in the standard deviations of HD indicating increased robustness towards variations in the test data. Further, iterating the teacher-student training yielded some performance gains, but only till the second iteration. A few representative examples from the results obtained by *basic teacher + robust student* are shown in Figure 2.

### 4.1. Comparison with the State-of-the-art (SOTA)

In comparison to SOTA approaches for CT image segmentation for OARs in cervical cancer radiation treatment (shown in Table 2), the performance of our approach seems better for the bowel bag, similar for the bladder and hips, but slightly worse for the rectum. Note that the results in (Wang et al., 2020; Liu et al., 2020b,a; Rigaud et al., 2021) correspond to a small test dataset resulting from a single random split, which is susceptible to bias introduced during the splitting of the data. In terms of test dataset size, a comparison with (Rhee et al., 2020) is more suitable. However, (Rhee et al., 2020) had a comparatively larger training dataset also and trained separate models for each OAR. We believe that using our approach in combination with the data from (Rhee et al., 2020) may result in a better performance with a single model.

## 5. Clinical Acceptability Test

We conducted a validation study to assess the clinical acceptability of the automatically generated OARs segmentations. We used the *basic teacher + robust student* model from the first data-split, to predict OARs segmentation masks in the first 4 scans in the test dataset, which were used to generate automatic contours. We showed[3] both the clinically available contours and the automatically generated contours to a radiation oncologist (henceforth referred to as 'clinical expert'), without informing them about the method used to generate the contours. The clinical expert graded each contour for its clinical acceptability according to a 4-point Likert scale: 1=acceptable as it is, 2=acceptable but marginally deviating from exact anatomical definition (subjective to an observer), 3=acceptable with minor corrections because either a part of the organ was not delineated or a peripheral tissue was included in the contour, 4=not acceptable because a correction involving both deletion, as well as delineation of an additional contour, was required.

---

3. The contours were presented on 2D transverse slices spaced at a 10mm distance to make it similar to the clinical scenario where the contours are delineated on 2D transverse slices. The clinical expert optionally inspected the contours and scans in coronal and sagittal view also to ensure comprehensiveness.

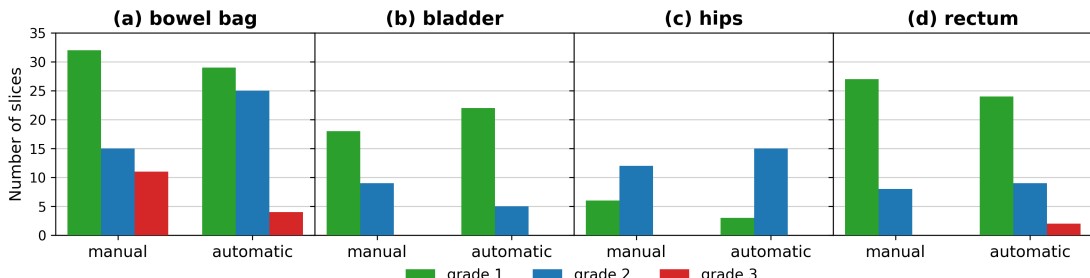

Figure 3: Comparison of clinical acceptability grades (smaller value indicates better quality) for clinically available contours (manual) and the contours generated from OARs segmentation masks predicted by our approach (automatic) for (a): bowel bag, (b): bladder, (c): hips, and (d) rectum.

The clinical acceptability grades for the automatically and manually generated contours for all the graded 2D transverse slices and OARs are shown in Figure 3. None of the contours were given grade 4 implying that all the contours were of clinically acceptable quality either as it is or with adaptations. Further, not all of the clinically available contours were graded as 1, representing inter-observer variation. A Chi-squared test of goodness of fit indicated that the histograms of clinical acceptability grades of the automatically generated contours were significantly different from the manually generated contours for the bowel bag $(\chi^2(1, N=58) = 11.402, p = 0.003)$. However, as shown in the Figure 3, it was unclear which contours (automatically or manually generated) were better. The clinical acceptability grades for automatically and manually generated contours were not significantly different for the bladder $(\chi^2(1, N=27) = 2.667, p = 0.102)$, and hips $(\chi^2(1, N=18) = 2.250, p = 0.134)$. For the rectum, the Chi-squared test statistics could not be obtained because the frequency counts corresponding to grade 3 were less than 5, however, it is apparent from the Figure 3 that the frequency counts in each category were similar for both the automatically and manually generated contours.

Qualitatively, the differences in grade 1 and grade 2 in all the organs were mainly attributed to inter-observer variance. In the case of hips, the window width and window level settings used to visualize the CT scans also influenced the difference between grade 1 and grade 2. Grade 3 corresponded to contours including mesorectum as a part of the bowel bag, and difference in cranial-caudal extent in the rectum.

## 6. Discussion and Conclusions

We investigated the possibility of using a large clinically available dataset of the abdominal region to learn a deep learning model for the automatic segmentation of OARs in cervical cancer radiation treatment. To the best of our knowledge, this is one of the few works in the direction of utilizing a large clinically available dataset containing missing annotations for learning a deep learning model. Our experimental results show that learning from a large dataset using our proposed approach yields significant performance gain despite missing annotations in the data. The obtained segmentations from our deep learning model were of clinically acceptable quality, which is encouraging.

Limitations of our work include an ablation study involving only a single run (i.e., network initialization), and a lack of experiments with different semantic segmentation architectures. Both decisions were consciously taken to find sensible results despite the expensive nature of training deep neural networks. Interesting future directions are 1) extending the current work to automatic segmentation of more OARs in cervical cancer radiation treatment e.g., sigmoid and anal canal, and 2) evaluating and learning from datasets of multiple hospitals and demographics to investigate and reduce possible bias in the predictions.

In conclusion, we demonstrated that training a deep learning model without using curated and specifically annotated medical imaging data, but with the capability of predicting clinically acceptable segmentation is possible. Apart from saving clinicians' time, our proposed approach leads to faster development time because of using the readily available data and increased test performance because of the increased dataset size.

## Acknowledgments

The research is part of the research programme, Open Technology Programme with project number 15586, which is financed by the Dutch Research Council (NWO), Elekta (Elekta AB, Stockholm, Sweden) and Xomnia (Xomnia B.V., Amsterdam, the Netherlands). Further, the work is co-funded by the public-private partnership allowance for top consortia for knowledge and innovation (TKIs) from the Ministry of Economic Affairs.

We thank Jan Wiersma (email: wiersmaj@amsterdamumc.nl), Jeroen de Vries (email: j.devries6@amsterdamumc.nl), and Bart van de Poel (email: bvandepoel@gmail.com) for their contributions in obtaining the data, data curation, and data cleaning, respectively, in the initial stage of the project.

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

## Appendix A. Description of Thresholds for Automatic Data Cleaning

The histograms of the cranial border of the scans, and the cranial border of the bowel bag annotation with respect to the most cranial point of the hip annotations in $D_f$ are shown in the Figure 4. The thresholds to crop the scans and delete the bowel bag annotations in the cranial direction are marked in the Figure.

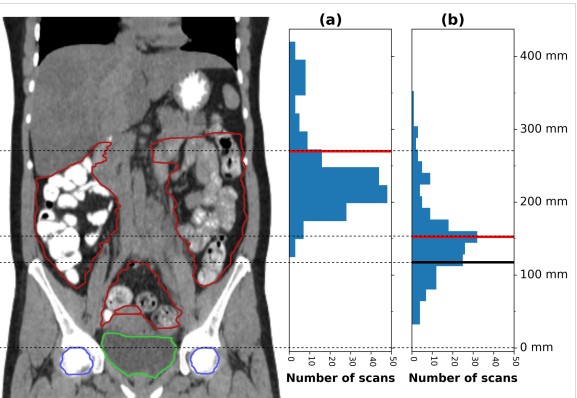

Figure 4: The histograms of the number of scans with respect to the distance from the most cranial point of the hip annotations to (a) the cranial border of the scan, and (b) the cranial border of the bowel bag annotation. On the left, a representative CT scan and reconstructed contours in the coronal view are shown (red: bowel bag, green: bladder, blue: hips). Red lines in (a) and (b): thresholds to crop the FOV and delete the bowel bag annotations in the cranial direction. The black line in (b): threshold for discarding the scans, where the bowel bag annotations did not cover the pelvic region. Dashed lines: corresponding anatomy for each threshold.

## Appendix B. Performance Metrics for all OARs

| Method | Bowel bag | Bladder | Hips | Rectum |
|---|---|---|---|---|
| 3D U-Net $\mathcal{D}_f$ | 79.47 (11.24) | 89.25 (16.05) | 91.57 (3.73) | 73.58 (13.93) |
| 3D U-Net $\mathcal{D}_f^{clean}$ | 84.77 (6.71) | 90.24 (15.62) | 91.91 (2.23) | 73.15 (15.32) |
| *basic teacher* | 84.88 (6.21) | 90.61 (14.59) | 91.81 (2.50) | 74.15 (14.05) |
| *basic student* | 86.31 (5.59) | 92.13 (11.43) | 92.65 (2.14) | 76.95 (12.63) |
| *robust teacher* | 84.69 (7.01) | 90.23 (15.43) | 91.73 (2.40) | 74.58 (12.70) |
| *basic teacher + robust student* | 85.86 (5.57) | 92.08 (10.03) | 92.62 (2.19) | 77.86 (11.96) |
| *robust teacher + robust student* | 86.25 (5.54) | 91.93 (10.58) | 92.34 (2.22) | 78.10 (10.99) |
| *robust teacher + robust student* - iter. 2 | 86.12 (5.50) | 92.39 (8.88) | 92.69 (2.16) | 78.39 (11.92) |
| *robust teacher + robust student* - iter. 3 | 86.40 (5.54) | 92.31 (7.60) | 92.76 (2.23) | 77.92 (12.68) |

Table 3: Mean (standard deviation) of Dice coefficient of the best models obtained from 5-fold cross-validation. Aug.: additional augmentations.

| Method | Bowel bag | Bladder | Hips | Rectum |
|---|---|---|---|---|
| 3D U-Net $\mathcal{D}_f$ | 61.55 (10.77) | 88.24 (16.99) | 96.62 (4.71) | 74.51 (14.99) |
| 3D U-Net $\mathcal{D}_f^{clean}$ | 66.37 (9.27) | 90.07 (16.36) | 97.03 (3.27) | 74.52 (15.51) |
| *basic teacher* | 66.45 (8.68) | 90.55 (15.61) | 96.86 (3.84) | 75.46 (14.10) |
| *basic student* | 68.91 (8.57) | 93.13 (11.97) | 97.55 (3.18) | 78.96 (12.92) |
| *robust teacher* | 66.09 (8.84) | 90.46 (16.25) | 96.80 (3.47) | 75.86 (12.59) |
| *basic teacher + robust student* | 68.33 (8.61) | 92.95 (10.43) | 97.53 (3.23) | 80.24 (12.10) |
| *robust teacher + robust student* | 68.97 (8.40) | 92.74 (10.78) | 97.29 (3.34) | 80.30 (11.37) |
| *robust teacher + robust student* - iter. 2 | 69.10 (8.26) | 93.57 (9.44) | 97.50 (3.21) | 81.01 (11.95) |
| *robust teacher + robust student* - iter. 3 | 69.58 (8.32) | 93.26 (8.55) | 97.52 (3.30) | 80.59 (12.61) |

Table 4: Mean (standard deviation) of Surface Dice computed at a tolerance of 2.5mm (voxel spacing) of the best models obtained from 5-fold cross-validation. Aug.: additional augmentations.

| Method | Bowel bag | Bladder | Hips | Rectum |
|---|---|---|---|---|
| 3D U-Net $\mathcal{D}_f$ | 35.27 (24.77) | 7.84 (10.51) | 4.16 (20.14) | 16.98 (11.86) |
| 3D U-Net $\mathcal{D}_f^{clean}$ | 19.34 (11.70) | 9.68 (37.38) | 2.93 (1.00) | 17.80 (13.26) |
| *basic teacher* | 18.43 (9.97) | 7.96 (26.75) | 2.95 (1.03) | 17.10 (12.61) |
| *basic student* | 17.26 (10.47) | 6.31 (22.70) | 2.87 (1.04) | 16.11 (18.35) |
| *robust teacher* | 18.43 (10.83) | 7.56 (22.30) | 2.95 (1.05) | 17.34 (17.86) |
| *basic teacher + robust student* | 17.55 (10.23) | 5.57 (15.12) | 2.87 (1.05) | 15.58 (18.00) |
| *robust teacher + robust student* | 17.10 (11.23) | 5.12 (7.22) | 2.90 (1.08) | 14.57 (10.63) |
| *robust teacher + robust student* - iter. 2 | 17.23 (10.74) | 4.71 (6.55) | 2.88 (1.08) | 14.58 (11.29) |
| *robust teacher + robust student* - iter. 3 | 17.41 (11.76) | 4.49 (5.22) | 2.88 (1.14) | 15.05 (11.94) |

Table 5: Mean (standard deviation) of Hausdorff distance at 95 percentile of the best models obtained from 5-fold cross-validation. Aug.: additional augmentations.

