# OpenReview forum: "Learning Clinically Acceptable Segmentation of Organs at Risk in Cervical Cancer Radiation Treatment from Clinically Available Annotations"
_MIDL.io/2023/Conference — MIDL 2023 Oral_

### Official Review · Reviewer_Fi2t · 2023-01-29

**Confidence:** 5
**Preliminary Rating:** 5
**Recommendation:** Oral, Poster

**Summary:**

This work is about automation of OAR segmentation in radiotherapy. This area has the specific advantage that contouring is part of the clinical practice, so it is possible to extract thousands of cases from PACS. However, actually doing so comes with a number of challenges, and the authors describe those and their approach of dealing with them, such as harmonization of structure labels and identifying a range of transversal slices for which the structures are likely to be reliable. There main methodological contribution is a teacher-student setup for imputing missing structures, including an uncertainty-based weighting of the pseudo labels. The evaluation is done using three sensible segmentation quality measures and statistical testing.

**Strengths:**

While the idea alone of using archived radiotherapy contours for training neural networks is not novel, this is a nice report on a successful application of the technology, and the manuscript gives readers a good impression of what kind of obstacles have to be overcome and how that can be achieved. The results are convincing and the authors have performed a good quantitative analysis using a reasonable set of measures and appropriate (far above average!) statistical methods. I like the fact that while the main manuscript only contains overall segmentation quality measures, the interesting separate information about the four segmented structures is given in the appendix. Overall, the manuscript is well-written and easy to read.

**Weaknesses:**

There are no really serious weaknesses, but I still want to point out a few issues and will try to order them by decreasing severity.
The technical setup in particular w.r.t reference data, preprocessing, and model training is not described precisely and completely enough to be able to reproduce it exactly. I found a few other sentences in the methods and results section also unclear (details below).
The method is only used for four structures in the pelvic region, and only on a medium-size dataset. IOW, about one thousand cases is a reasonable and decent size, but not particularly large. Also note that only 134 cases had all four structures annotated in the pelvic region.
Athough I stated that the paper contains a methodological part, it just takes one possible approach and that may not be the optimal one. That approach is thoroughly evaluated, however, and the authors also reason well about a few decisions they made, so one could also have called it a decent application paper (NB: I think I can't see which category the authors chose).

**Deanonymize Review:**

yes

**Detailed Comments:**

In the "Data" section, you describe well the source and structure of your training data. The test data, however, which is at least as important, does not seem to be described at all. The sentence "In addition, we used 105 CT scans from 95 cervical cancer patients who received radiation treatment as test data." leaves open whether its an open or internal dataset, whether it's from the same hospital, whether it's in any way related to the training data, and what is the quality of the reference segmentation.

The preprocessing lacks details. For instance, according to section 2.1, HU are "converted to intensity values between 0 and 1 by windowing", but from which range? Section 3.3: Are you really using an unmodified original, 2D U-net architecture? "global brightness and contrast variations, random rotations" used for data augmentation is also quite unspecific (linear or nonlinear brightness? which angles? which axes?) Note that from my POV, you could save space in section 3.3 by removing a few very well-known references and the duplicate "network initialization" part. Another underspecified approach is "masking an organ with a random intensity" (section 4). The footnote indicating that the implementation will be made public could of course very well address the reproducibility problem, but I would still put a few more details into the manuscript.

Could you double-check the two sentences "Adding noise to the data through additional augmentations provided a marginal gain in the mean performance of the student model. Intuitively, additional augmentations provided a considerable decrease in the standard deviations of performance metrics indicating robustness towards variations in the test data."? The meaning and point of the second one was a little unclear to me, maybe just because it starts with "intuitively", and because its tone seems to contradict the first sentence. Maybe the two statements could be put into relation, if they both talk about the effects of the same augmentation and if you wanted to say that the mean performance has only improved very little, but the stddev has dropped considerably?

**Paper Type:**

both

**Questions To Address In The Rebuttal:**

I think the most important point of my above critics is the reference data description, which should be added.

I would also like the manuscript to be cleared up in the areas I pointed out in detail above during the revision phase.

---

### Official Review · Reviewer_JjwF · 2023-02-01

**Confidence:** 4
**Preliminary Rating:** 4
**Recommendation:** Poster

**Summary:**

The authors takle the problem of learning segmentations from sparsely annotated data, where only a few of the many studies have all organs of interest annotated, while the remainder only has annotations for a subset of the organs. In this setting, they propose to employ an approach where a first network is trained only on the few exhaustively annotated cases, then predicts all organs on all other cases. Through "annotation imputation" the existing and predicted labels are combined, so that a dataset is obtained where for all scans all organs are annotated. On this dataset, a second network is trained from scratch, taking into consideration an uncertainty estimate obtained from the first network's prediction. They dub the first network the "teacher", and the final network the "student".

**Strengths:**

* The paper addresses a common problem in medical image segmentation: Achieving a usable predictor from little and potentially noisy data.
* I like the evaluation scheme, where not some theoretical metric is put up front, but the relevant clinical question: is it good enough to reduce clinicians' workload in practice?
* 5-fold CV is seldom seen in deep learning experiments -- that's remarkable, as well as the thorough statistical comparison.

**Weaknesses:**

* Main objection to the paper for me is that it does not justify the choice of approach by more than an ablation study. It lacks an argument or experiment answering why the proposed approach is superior to ... what? The authors don't reflect on existing methods to solve the task, therefore all presented validation only very weakly suggests the necessity of the approach.
* In student-teacher setups, you frequently find iterative approaches (co-training with EMA, for example). Why was this not done here, but rather a simple one-off pipeline (that, in my humble opinion, isn't even something I'd call a student-teacher setup but just a re-training on an automatically generated label set)
* One fundamental question (from the causality perspective) isn't tackled at all: can unlabeled data in the causal setting of this problem theoretically help the task at all? See "Causality Matters in Medical Imaging", Castro 2020. I think this would provide a good foundation to discuss the results, though of course this is no "must-have" and a much smaller concern compared to the before.
* I find that the "heuristic for data cleaning" that is specific to the training dataset should not be elevated to one of three "key contributions". Doing so diminishes the more noteworthy results and even may make readers think that the other results aren't enough so that something had to be added to make it look fuller. (It's even the first key contribution in the list...)
* The important question of controlling for bias that the computer-generated labels surely introduce is not posed or explicitly or implicitly answered.
* The clinical evaluation for me lacks a focus on some questions I'd have had: a breakdown of the organs that are judged differently on the "usability" scale. Is this for one structure more often the case? What is the visual commonality of the less acceptable segmentations? How does this relate to the potential bias in the proposed pipeline? How could the heuristics be modified to counteract this? Etc.... Somehow, the paper ends where the methodological questions start.
* Sec. 5.: It's not "allowed" to select the best of cross-validated models. CV is the (less biased) estimator for the true model performance. One of the folds is a biased estimator. CV cannot be used for model selection.

**Deanonymize Review:**

yes

**Detailed Comments:**

* In 3.3. "network initialization" is duplicated.


**Paper Type:**

both

**Questions To Address In The Rebuttal:**

* I wondered about the selection of references to motivate an approach to "learning from partially labeled data". Alphafold has nothing to do with the segmentation task, and the entire setup and approach is completely different, and the conference paper on ImageNet classification isn't a very strong or scene-defining ref. either, and also neither from a comparable domain nor a similar task. I would have expected more targeted references on semi-supervised learning, few-shot learning, or similar, or review papers of which there are some. Also the host of literature on "learning from partially labeled data" is largely not reflected or cited, and it is therefore not sufficiently motivated why a new approach is needed.
* In Table 1, please add row-wise sums of how many total GT labels are available per structure. Just to complete the picture.
* It is neither referenced nor described what a K-head U-Net looks like: what is the head that is k-fold multiplicated? Is there a reference?
* Why does the student network also use 5 heads and therefore predict an uncertain map? And how is this further treated? Do you sample from the implicit distribution? Or do you average? And why at all?
* Table 2 is rather uninformative. I suggest to split up according to the textual description in the section: four models in the first section, three in the second. Also, the composition of model, augmentation, datasets is probably much easier to capture if these are "boolean" columns with check marks if it applies, instead of 3-line textual descriptions like "K-head student + Df (...) + Aug in teacher and student (...)". In addition, four significant digits are terrible to comprehend. If they were needed to show the effect (and they are not), then the effect would have been too small. But since there is significance in differences according to the daggers/asterisks, you'll want to make them visible. My feeling is that this table represents the core quantitative results. It should hence provide them to the reader in the greatest clarity possible -- and does exactly the contrary.
* There is no mention where the label in annotation imputation (Figure 1 b) comes from -- I suppose this is the mask of the known OaRs? Should be clarified. There is no description what the "annotation imputation" does to the teacher-predicted mask based on the "label". Needs to be mentioned.
* The histograms (or the description of the figure) could also profit from a do-over. Why not label the categories with short meaningful words/phrases instead of "grade 1" to "grade 3"? Acceptable "As is" - "Minor Mod." - "Sign. Mod.". (Subjective comment: I don't like the colors of the bars.)
* How have the 4 cases for the validation been selected, and from which part of the data?

In addition to these points, the remarks in "weaknesses" should be addressed as much as possible.

---

### Official Review · Reviewer_uUKJ · 2023-02-06

**Confidence:** 5
**Preliminary Rating:** 3

**Summary:**

This paper describes an approach where the authors developed a training strategy that utilized a teacher-student model, annotation imputation, and uncertainty guidance to produce segmentation of Organs At Risk on CT images for cervical cancer radiotherapy.
A dataset of 1170 scans from 1108 patients was used for training and validation and a separate test of 105 CT scans from 95 cervical cancer patients was used as a test set. The authors present relatively simple heuristics to clean the dataset and show that the training on the cleaned set leads to substantial improved performance. Table 1 shows that adding uncertainty guidance adds a small performance gain. A qualitative experiment was performed using 4 CT scans, which showed clinically acceptable OAR segmentations.

**Strengths:**

- Relatively large dataset of CT images
- Relevant clinical problem
- This study has clear practical tips for training a well-performing segmentation model for OAR segmentation
- Clear ablation experiments showing the benefit of the various aspects of this work

**Weaknesses:**

- This is a 3D problem, but the authors used the 2D U-Net method. It is unclear how the 2D results are postprocessed in a 3D segmentation. Or is the full validation done in 2D per slice?
- No comparison to previous state-of-the-art segmentation approaches for OAR segmentation. For example this paper : https://arxiv.org/abs/1809.04430. How does the current work compare to this?
- No novelty in the methodological aspects of this work.
- The authors claim to use a self-supervised learning method, but this is semi-supervised learning.
- The automatic segmentation of the hips worked not so well, seems inferior to manual segmentation, but no discussion why this is the case.

**Deanonymize Review:**

no

**Detailed Comments:**

The authors use the term self-supervised training approach, but this is semi-supervised learning since a limited set of actual image labels is used.

**Paper Type:**

both

**Questions To Address In The Rebuttal:**

- Please provide a clear comparison to previous work in the field.
- Please explain how a 2D U-Net is used for this 3D task. Or is everything done in 2D?
- Please include a discussion on the seemingly worse perforrmance for the hips.

---

### Meta-Review · Area_Chair_bVex · 2023-02-25

**Recommendation:** Accept (Poster)
**Confidence:** 5

**Metareview:**

Clear concensus by reviewers that this work is of sufficient quality and substance to be published at MIDL.

The work looks into OAR segmentation in radiotherapy, and strategies for teaching a model with imperfect labels, using a student-teacher framework. The work brings together a number of techniques to achieve good performance in the task. Although the individual components may not be technically novel, the work brings them all together into a nice framework to show promising results in this task, which should be of interest to a part of the community.